# Engineering and optimising deaminase fusions for genome editing

Luhan Yang[1,2,3,†], Adrian W. Briggs[1,*], Wei Leong Chew[1,2,*,†], Prashant Mali[1], Marc Guell[1,3,†], John Aach[1], Daniel Bryan Goodman[1,4], David Cox[4], Yinan Kan[1,3,†], Emal Lesha[1,3,†], Venkataramanan Soundararajan[1], Feng Zhang[5,6,7] & George Church[1,4,8]

Precise editing is essential for biomedical research and gene therapy. Yet, homology-directed genome modification is limited by the requirements for genomic lesions, homology donors and the endogenous DNA repair machinery. Here we engineered programmable cytidine deaminases and test if we could introduce site-specific cytidine to thymidine transitions in the absence of targeted genomic lesions. Our programmable deaminases effectively convert specific cytidines to thymidines with 13% efficiency in Escherichia coli and 2.5% in human cells. However, off-target deaminations were detected more than 150 bp away from the target site. Moreover, whole genome sequencing revealed that edited bacterial cells did not harbour chromosomal abnormalities but demonstrated elevated global cytidine deamination at deaminase intrinsic binding sites. Therefore programmable deaminases represent a promising genome editing tool in prokaryotes and eukaryotes. Future engineering is required to overcome the processivity and the intrinsic DNA binding affinity of deaminases for safer therapeutic applications.

[1] Department of Genetics, Harvard Medical School, Boston, Massachusetts 02115, USA. [2] Program in Biological and Biomedical Sciences, Harvard Medical School, Boston, Massachusetts 02115, USA. [3] eGenesis Inc., 1 Kendal Square, Building 200, Cambridge Biolabs, Cambridge, Massachusetts 02139, USA. [4] Harvard-MIT Division of Health Science and Technology, Cambridge, Massachusetts 02139, USA. [5] Broad Institute of MIT and Harvard, Cambridge, Massachusetts 02142, USA. [6] McGovern Institute for Brain Research, MIT, Cambridge, Massachusetts 02139, USA. [7] Department of Brain and Cognitive Sciences, MIT Cambridge, Cambridge, Massachusetts 02139, USA. [8] Wyss Institute for Biologically Inspired Engineering, Boston, Massachusetts 02115, USA. † Present addresses: eGenesis Inc., 1 Kendall Square, Building 200, Cambridge Biolabs, Cambridge, Massachusetts 02139, USA (L.Y., M.G., Y.K. and E.L.); Genome Institute of Singapore, Agency for Science, Technology and Research (A*STAR), Singapore 138672, Singapore (W.L.C). * These authors contributed equally to this work. Correspondence and requests for materials should be addressed to L.Y. (email: luhan.yang@eGenesisbio.com) or to G.C. (email: gchurch@genetics.med.harvard.edu).

Genetic modification of mammalian cells has been greatly facilitated by the development of customized zinc finger (ZF)[1,2], transcription activator-like effectors (TALEs)[3] nucleases[4,5] and clustered regularly interspaced short palindromic repeat/Cas9 (CRISPR/Cas9 (refs 6–8). These programmable nucleases create targeted genomic double-strand breaks (DSBs) that enhance gene conversion from exogenous homology donors via homology-dependent repair (HDR). However, HDR faces numerous limitations: First, the nuclease-induced DSBs are associated with genomic aberrations and cytotoxicity[9], which is more concerning when targeting multiple genomic loci[10]. Second, HDR is highly inefficient compared with the competing non-homologous end joining pathway[11,12], which results in generation of random insertions/deletions (indels) instead of the intended genetic modifications. Third, *in vivo* genome editing remains challenging due to the difficulty in delivering donor DNA at sufficient concentration and the low HDR activity in somatic cells[13].

Deaminases are naturally occurring proteins that operate in various important cellular processes. Activation induced deaminase (AID) and apolipoprotein B mRNA editing enzyme catalytic polypeptide-like family proteins (APOBECs)[14] are cytidine deaminases critical to antibody diversification and innate immunity against retroviruses[14]. These enzymes convert cytidines (C) to uracils (U) in DNA. If DNA replication occurs before uracil repair, the replication machinery will treat the uracil as thymine (T), leading to a C:G to T:A base pair conversion[15]. This elegant editing mechanism suggests a simple and effective genome editing tool that might help us circumvent the limitations associated with nuclease-based approaches.

Recently, programmable deamianses by fusing APOBECs with catalytically dead and nicking Cas9 have been reported[16]. This finding holds great promise for therapeutic editing due to its high on-target efficiency and low indel rate. However, its tendency to induce off-target deamination has not been characterized systematically. Here, we aim to test if we could modulate the specific activities of chimeric deaminases and examine its specificity both near the target site and in the global level.

Here, we demonstrated that programmable deaminases could be generated by fusing cytidine deaminases with the ZF or TALE-DNA binding modules. They could site-specifically convert cytidines to thymidines with 13% efficiency in *Escherichia coli* and 2.5% in human cells under optimized conditions. However, off-target deaminations were detected both near the target sites and genome-wide, indicating significant enzyme processivity and off-target activity. Future engineering is required to increase the specificity of programmable deaminases for therapeutic applications.

## Results

### Design of targeted deaminases

To test this, we first engineered targeted deaminases by fusing each candidate deaminase (human APOBEC1, APOBEC3F, APOBEC3G (2K3A) (ref. 17) and AID) with a sequence-specific ZF, which has proven to be effective by previous studies (recognizing the 9bp DNA sequence 5′-GCCGCAGTG-3′ (ref. 18) (Fig. 1a). On the basis of available structures of APOBEC2, we tethered the ZF to the N-terminus of the deaminases to prevent steric hindrance to catalysis, separated by a four amino-acid linker (Fig. 1c). To determine editing efficiency *in vivo*, we integrated a single-copy GFP reporter into the *E. coli* genome[19] (Fig. 1b and Methods) in which the GFP is normally not expressed due to a 'broken' start codon ('ACG') and the ZF binding site is 9 bp from the target 'C' in the start codon. Correction of the genomic ACG to ATG by targeted deamination would restore GFP protein expression, thereby producing

GFP-positive cells quantifiable by flow cytometry. Among the four deaminase domains we tested, AID induced the highest GFP correction at the ACG site (Fig. 1c), thus the ZF-AID fusion was used for subsequent characterizations. No signal was observed in the control whereby only the ZF domain was expressed, indicating that GFP expression is attributed to the deaminase activity. We confirmed the intended ACG-to-ATG conversion in 20/20 randomly chosen GFP+ bacterial colonies from the ZF-AID fusion, as assessed by Sanger sequencing. Therefore, ZF-AID introduces C-to-T mutations at the locus specified by the fused DNA-binding module. We do not exclude the potential activities of other deaminase domains as ACG is not in the preferred deaminase sites for some of them.

We next asked if we could reprogram the targeting site of AID with different DNA binding module. To achieve this, we engineered a TALE-AID fusion (recognizing a different binding sequence 5′-TCACGATTCTTCCC-3′ (ref. 20) as reported in previous studies) and a corresponding reporter *E. coli* strain (Fig. 2c). Induction of TALE-AID for 10 h led to GFP expression in 0.02% of the reporter population, lower than that from ZF-AID, but nonetheless significantly higher than with TALE or AID expression alone (t-test, two-tailed, $P_{(TALE-AID, TALE)} = 0.0069$, $P_{(TALE-AID, AID)} = 0.0186$; $n = 4$) (Fig. 1d). Importantly, GFP expression is dependent on correct sequence recognition, because TALE-AID and ZF-AID do not induce GFP expression in reporter *E. coli* cells lacking the cognate target sequences (Fig. 1d). Additional target DNA sequences do not increase editing efficiency, suggesting that a single ZF-AID or TALE-AID is sufficient for editing (Supplementary Fig. 2 and Supplementary Note 8). Thus, ZF-AID and TALE-AID converts C-to-T at sequence-defined genomic loci.

The results demonstrated feasibility of using programmed deaminases for genome editing, but editing efficiency was low. We reasoned that the endogenous uracil repair pathways could reverse the targeted deamination, which would limit the desired C-to-T conversion. Therefore, we knocked out *mutS* and *ung*, two genes critical for uracil repair. Editing by ZF-AID increased to 0.5% (five-fold) in the Δ*mutS* knockout, and to 3.5% (35-fold) in the Δ*mutS* Δ*ung* double knockout (Fig. 1e). Similarly, editing by TALE-AID increased to 0.1% (seven-fold increase) in the Δ*mutS* Δ*ung* knockout (Fig. 1e). We confirmed GFP fluorescence signal by microscopy (Fig. 1f) and confirmed C-to-T transitions by sequencing the *gfp* gene of 20 randomly chosen GFP+ colonies from both the ZF-AID- and TALE-AID-induced populations. Hence, suppression of uracil repair, as well as the mismatch repair pathways increases editing frequencies from the targeted deaminases. All subsequent experiments in *E.coli* were done in the Δ*mutS* Δ*ung* background.

### Optimization of targeted deaminases

We next conducted structural optimization of the targeted deaminases by varying linker lengths and sequence compositions[21,22] (Fig. 2a). While tested variants all led to robust GFP rescue, with ZF-8-aa-AID achieving 7.5% GFP+ frequency after 10 h (Fig. 2b) and 13% after 30 h of induction (Supplementary Fig. 3 and Supplementary Note 9). Sequence composition of the linker also influences editing frequencies (t-test, two tailed, $P = 0.0032$ and $n = 4$). Hence, the linker determines performance of the overall construct. Moreover, both the ZF binding[23] and AID deamination are associated with the transcriptional activity of the target locus[24–28]. We manipulated the transcription status of GFP during the targeted deaminase expression. Consistent with previous observations, our results suggest that targeted deaminases work more efficiently if the target locus is transcriptionally active (Supplementary Fig. 4 and Supplementary Note 10).

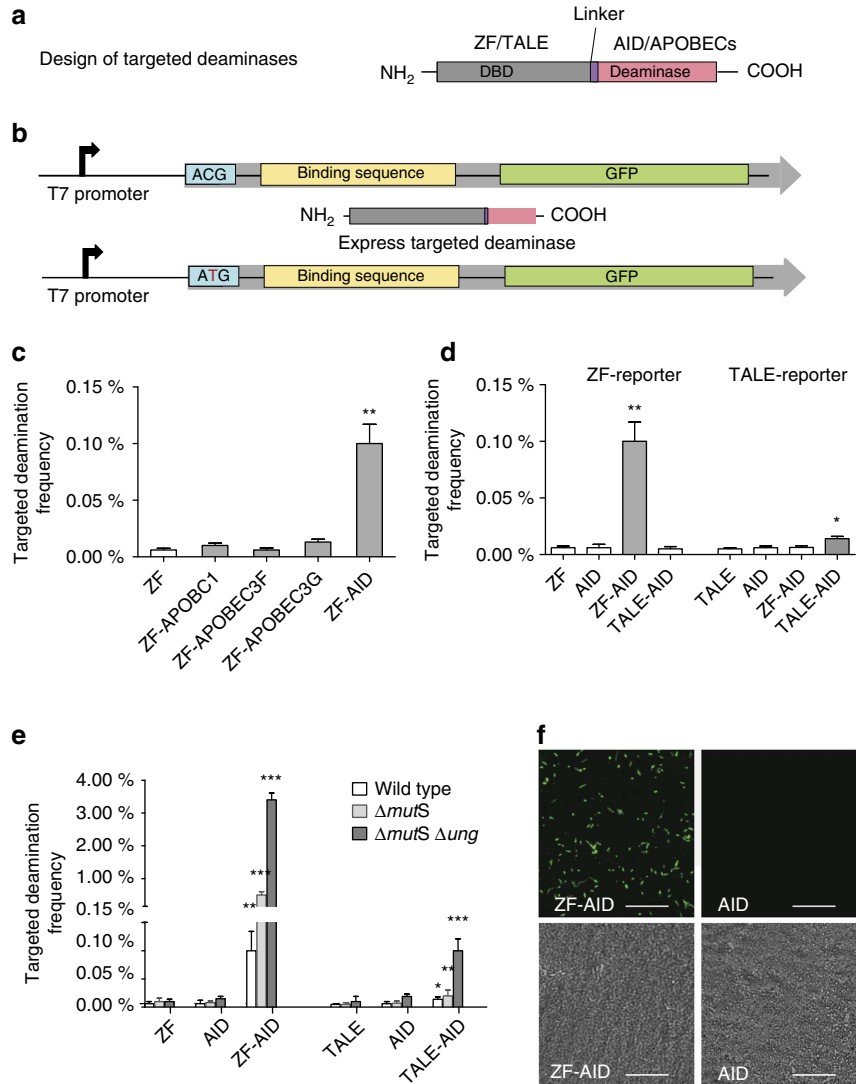

**Figure 1 | Design and targeted deaminase activity of chimeric deaminases in *E.coli*.** (**a**) Schematic representation of the design of targeted deaminases. The DNA binding domain (DBD), either ZF or TALE, was fused to the N-terminus of the deaminase with a certain linker. (**b**) Experimental overview: we integrated a GFP cassette (top) consisting of a broken start codon ACG, DNA binding sequence and the GFP coding sequence into the bacterial genome. We subsequently transformed targeted deaminases (middle) in pTrc-kan plasmid into the strain and induced protein expression. Targeted deamination of the C in the broken start codon leads to a ACG→ATG transition (bottom), rescuing GFP translation which is quantifiable via flow cytometry. (**c**) ZF-deaminases were tested for targeted deaminase activity by measuring GFP rescue. ZF, ZF-APOBECs (ZF-APOBEC1, ZF-APOBEC3F, ZF-APOBEC3G) or ZF-AID indicate cells transformed with plasmids that express ZF, ZF-APOBECs or ZF-AID respectively. All error bars indicate s.d. (All *t*-tests compare ZF-deaminases against the ZF control. *$P_{value} < 0.05$, **$P_{value} < 0.01$, ***$P_{value} < 0.001$, $n = 4$). (**d**) GFP rescue by ZF-AID and TALE-AID in the ZF-reporter and TALE-reporter strains.(All *t*-tests compare the fusion deaminases against the AID control. *$P_{value} < 0.05$, **$P_{value} < 0.01$, ***$P_{value} < 0.001$, $n = 4$). (**e**) GFP rescue by ZF-AIDs and TALE-AID in (wild type), (Δ*ung*), and (Δ*mut*S Δ*ung*) strains. All error bars indicate s.d. (All *t*-tests compare the fusion deaminases against the AID control. *$P_{value} < 0.05$, **$P_{value} < 0.01$, ***$P_{value} < 0.001$, $n = 4$). (**f**) *E.coli* (Δ*mut*S Δ*ung*) cells imaged under fluorescence (upper) and phase contrast (lower) after expression of ZF-AID or AID for 10 h. Top, Scale bar, 20 μm. More detailed structures and sequences of the fusion proteins and reporters are shown in Fig. 2a,c, Supplementary Fig. 1 and Supplementary Notes 1–7.

Our initial TALE –AID (hereafter referred to as TALE-C1-AID) is less efficient than the ZF-AIDs (Fig. 1e). Given the importance of the linker between the DNA-binding module and the deaminase, also considering that the TALE-C terminus was engineered as a linker for many TALE fusion proteins, we proceeded to investigate if truncation of the 178aa (ref. 20). C-terminus of TALE could increase TALE-AID activity (Fig. 2c). Truncations were chosen at *in silico* predicted loop regions. We also constructed five bacterial GFP reporter strains, each with a genomic *gfp* locus carrying a broken start codon 2, 5, 8, 11 or 14 bp upstream of the TALE binding site (Fig. 2c). Targeted deamination frequencies were then measured by GFP rescue

frequency and compared in a 5-by-5 matrix of TALE-AIDs and reporters (Fig. 2d). TALE-AID truncations showed significantly higher GFP rescue over that of TALE-C1-AID (Fig. 2d), with TALE-C3-AID achieving a genomic editing frequency of 2.5% on the 8 bp-spacer reporter after 10 h of induction (Fig. 2d), and 8% following 20 h of induction (Supplementary Fig. 3). Interestingly, TALE-C3-AID outperformed all other constructs regardless of the reporter spacer length, suggesting that this chimeric protein has an intrinsically optimal structure out of the TALE-AIDs tested. These results for ZF-AIDs and TALE-AIDs thus reveal important design considerations for engineering efficient targeted deaminases (Fig. 2d).

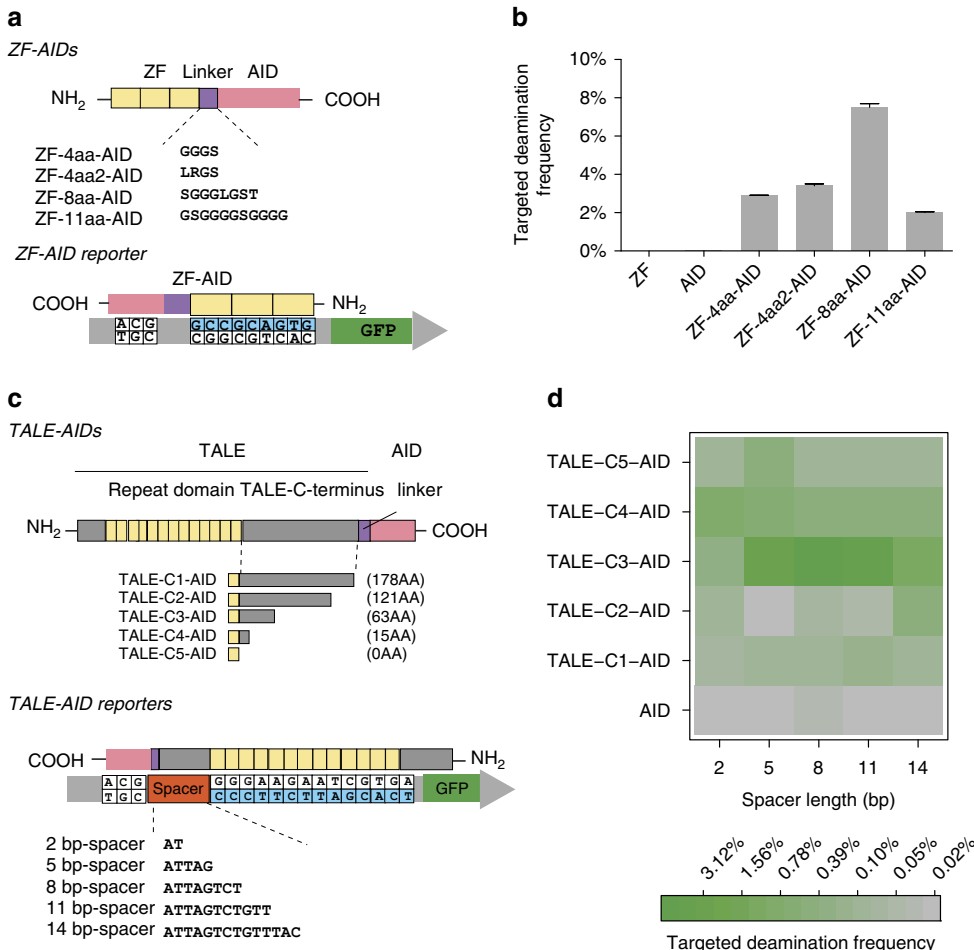

**Figure 2 | Optimization of targeted deamination frequency of AID fusions in *E.coli*.** (**a**) Schematic representation of ZF-AIDs variants tested for targeted deaminase activity (upper) and the reporter (lower) with the ZF-recognition sequence in blue. (**b**) GFP rescue by expression of the four ZF-AIDs variants and ZF or AID domains alone. All error bars indicate s.d. (**c**) Schematic representation of TALE-AIDs and the reporters tested for targeted deaminase activity. Five TALE-AIDs (upper) with different TALE C-terminus truncations (C1–C5) were constructed, with the remaining C-terminus lengths shown in parentheses. Full TALE-AID protein sequences can be found in Supplementary Note 2. Five reporters were constructed (lower) with different spacer lengths (2, 5, 8 and 11 bp) between the broken start codon and TALE DNA binding motif. The TALE binding site on the GFP reporter is shown in blue; the TALE N-terminus segment specifies the 5′ thymine base of the binding site. (**d**) All five TALE-AIDs were tested for targeted deaminase activity on all five reporters (**c**). Green and grey encode high and low GFP rescue, respectively.

**Specificity of targeted deaminases**. Having investigated and improved deaminase targeting frequency, we next characterized targeting specificity of programmable deaminases from three aspects: (1) to test the stringency of the sequence requirement, we investigated the C-to-T modification efficiencies at sequences similar to DNA recognition sites; (2) to assess the accuracy of C-to-T modification near DNA binding sites, we deep sequenced the GFP locus from a population of targeted deaminases-treated bacteria; and (3) to unbiasedly assess the specificity, we conducted whole genome sequencing of modified cells.

First, we tested the genome editing efficiency of programmable deaminases in *E.coli* reporter cells caring reporters with a mutated DNA recognition sequence. We observed that single-nucleotide change in the ZF-cognate target sequence led to 4–8 fold decrease in observed editing rates (Fig. 3a), indicating that ZFP-8aa-AID is specific to the target locus. We next investigated the specificity of TALE-AID by individually varying each nucleotide in the TALE recognition site to the second most preferred base[1] for that position (Fig. 3b), and tested TALE-AID targeting efficiency on individual reporters respectively. Interestingly, TALE-C3-AID, which was designed to recognize a 14 bp sequence, showed strong sequence specificity only for the first 9 bp proximal to the target

site (5′ CTTCTTCCC 3′ in the TALE recognition site). For reasons that remain to be investigated, sequence alterations at more distal positions in the TALE binding site led to variable targeting frequencies (Fig. 3b).

Next, to test if programmable deaminase can pinpoint the deamination activity to designated C nearby DNA binding sites, we sorted 10,000 GFP + and 10,000 GFP − cells after 30 h of ZF-8aa-AID induction, and randomly isolated 200 individual colonies from each population. We Sanger sequenced 1 kb surrounding the *gfp* target site and, as control, the constitutively expressed *gapA* gene, which lies 1.9 Mbp away from *gfp*. In the GFP + population, all colonies harboured the intended C-to-T transition in the *gfp* start codon. Interestingly, 5.5% (11/200 colonies) of these colonies carried additional C-to-T transitions in the GFP transgene (Fig. 3c). Most of these additional mutations were confined in a ± 15 bp region flanking the ZF binding site, mutations > 150 bp away were also detected, suggesting that catalytically processivity of AID (ref. 29) is retained in the engineered protein even with ZF to anchor it at the targeting site. In the GFP − population, the only variant detected over 200 colonies was a C-to-A transition 1 bp away from the intended target site (ACG-to-ACA) that is present in 2% of the population

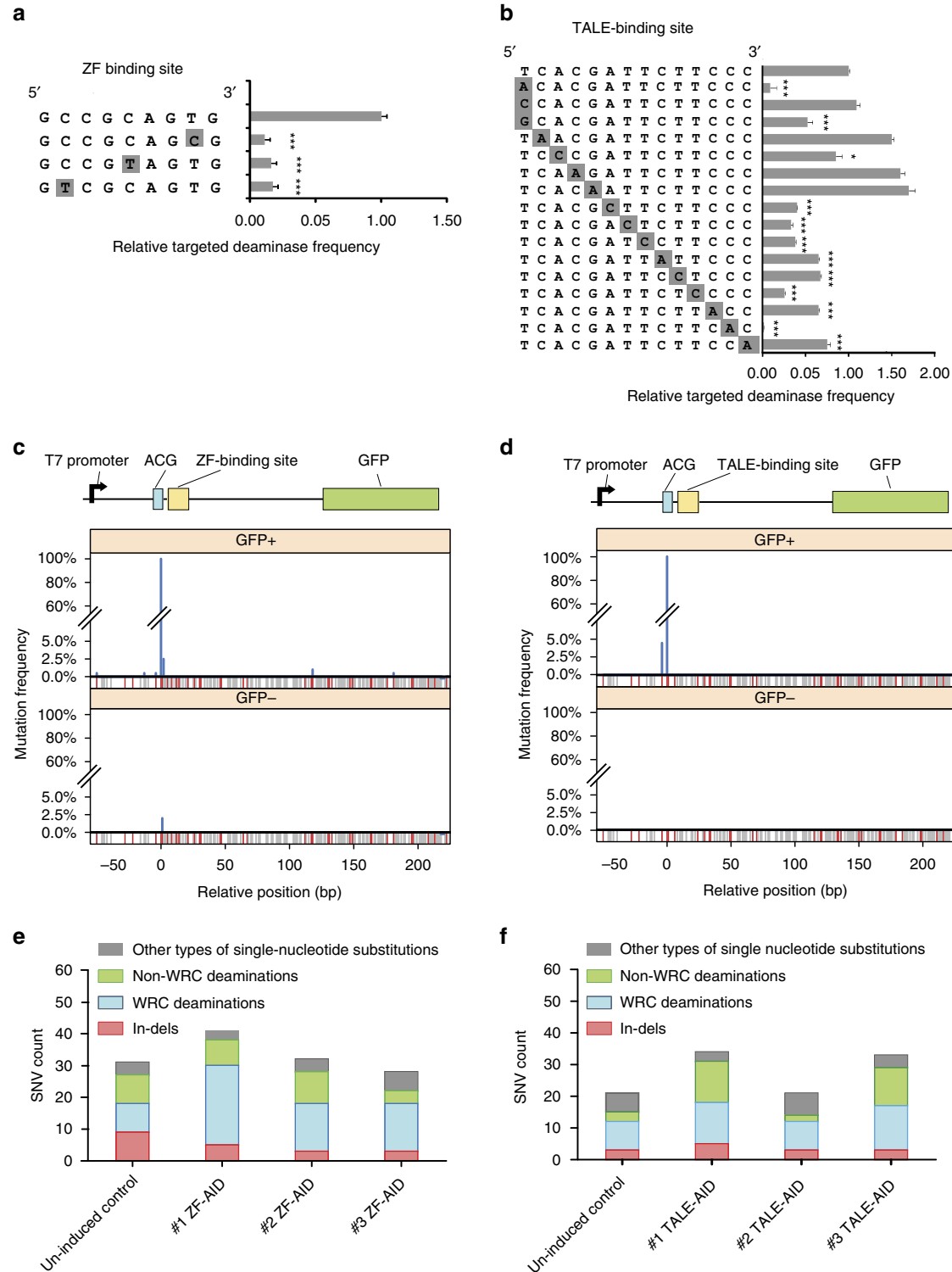

**Figure 3 | Test of the specificity of AID fusions.** (**a**) Test of ZF-8aa-AID sequence specificity using a GFP reporter with point-mutated ZF binding sequences. *t*-tests compare each mutated site against the unmodified site (top). *$P_{value}$ < 0.05, **$P_{value}$ < 0.01, ***$P_{value}$ < 0.001, n = 4. All error bars indicate s.d. (**b**) Test of TALE-C1-AID sequence specificity using a GFP reporter with point-mutated TALE binding sites. *t*-tests compare each mutated site against the unmodified site (top). *$P_{value}$ < 0.05, **$P_{value}$ < 0.01, ***$P_{value}$ < 0.001, n = 4. All error bars indicate s.d. Note that we altered the first nucleotide, a TALE-N terminus-specified thymine, to three other nucleotides individually, while we changed other nucleotides in the TALE recognition domain to the nucleotide mostly likely to be recognized[3]. (**c**) Mutation location and spectrum in the GFP gene of GFP + and GFP − cells collected after ZF-8aa-AID induction. A schematic structure of the GFP gene is shown above the mutation frequency along the gene's length among 200 Sanger sequenced colonies of each cell population. Grey lines indicate positions of C/G nucleotides; red lines indicate occurrences of the AID preferred motif (WRC). (**d**) Mutation spectrum on the GFP gene of GFP + and GFP − cells collected after TALE-C1-AID induction. (**e**) Whole-genome SNV profiles of strains with/without ZF-AID induction. SNVs that may stem from cytosine deamination (C/G→T/A) are in either green (if C was in the AID-preferred WRC motif) or blue (all other Cs) bars. (**f**) Whole-genome SNVs profiles of strains with/without TALE-AID induction. Colour schematic is the same as **e**.

(Fig. 3c). No mutations were found in *gapA* in any colony from the two populations. We next repeated our assay using TALE-C3-AID. In the GFP + population, besides the intended C→T mutation, an additional C→T mutation 4bp upstream of the intended site was found in 4.5% of the population (9/200 colonies, Fig. 3d). No other off-target mutations were detected in the GFP coding sequence or in the GFP − cells. We concluded that targeted deaminases have residual processivity that can mutate nearby C's within a ± 15 bp window of the target DNA sequence.

Finally, to assess genome-wide off-targeting, we sequenced with ∼50× coverage the genomes of three GFP + colonies edited by ZF-8aa-AID, and three colonies edited by TALE-C1-AID, and compared them to control GFP- colonies in which the expression of deaminases had not been induced. We did not observe increased indel mutations in the ZF/TALE-AID expressing clones (Fig. 3e,f, Wilcoxon, $P_{value} = 0.7109$). However, we detected elevated levels of genome-wide C:G→T:A transitions in WRC (W = A/T, R = A/G) sequence motifs following expression of targeted deaminases (Wilcoxon, $P_{value} = 0.02$, *t*-test) (Fig. 3e,f). In addition, we did not find any off-target mutations at predicted ZF/TALE off-target sites. The fact that off-target mutations are enriched at WRC motifs - the canonical AID recognition sequence[30] suggest that AID in the fusion protein still maintains its intrinsic DNA binding activities and contribute to elevated mutagenesis in the genome.

**Human genome engineering using targeted deaminases.** Given the intense interest in precise genomic editing for human biomedical studies, we tested functionality of our targeted deaminases in human cells. We constructed a mammalian reporter in which an EF1α promoter drives expression of a GFP harboring a broken-start-codon (ACG), followed by an IRES-mCherry selection marker. The reporter construct was stably integrated into HEK293FT cells by lentiviral transduction and a clonal cell line was isolated by FACS sorting (Fig. 4a). The optimized ZF-AID construct (ZF-8aa-AID) was then delivered into the reporter cell line via transfection (Fig. 4a). Following 48 h of ZF-8aa-AID expression, 0.12% of transfected cells turned GFP + . We next constructed ZF-AID$^{\Delta NES}$ by truncating the 15aa from the C-terminus of AID, which contains a strong nuclear export signal[31] and regions that interact with mismatch repair proteins[32]. This mutation, despite the potential to destabilize the protein, is also expected to correctly localize the ZF-AID to the nucleus and decouple AID from the mismatch repair pathway. The expression of ZF-AID$^{\Delta NES}$ significantly increased GFP + cell frequency compared with full-length ZF-AID (Fig. 4b) (0.56%, *t*-test, two-tailed, $n = 4$ and $P_{value} = 0.0013$). Encouraged by our *E.coli* study, we examined if inhibiting the counteracting pathways of uracil repair and mismatch repair would increase C-to-T transition in human cells. Interestingly, the combination of the UNG inhibitor UGI (ref. 33) and MSH2 shRNA increased ZF-AID$^{\Delta NES}$–mediated editing efficiency to 2.5% (Fig. 4b). In contrast, the expression of ZF$_{GFPINL}$-AID$^{\Delta NES}$, a fusion protein whose zinc finger domain targets a site 265 bp away from the GFP start codon, resulted in minimal GFP rescue (Fig. 4a,c), suggesting that genome editing by ZF-AID$^{\Delta NES}$ is sequence-specific. Successful C:G→T:A targeting of the broken start codon was confirmed by Sanger sequencing of the *GFP* locus in 8/8 stable GFP + colonies. Therefore, engineered deaminases are capable of efficient sequence-specific genome editing in HEK293FT cells, and editing efficiency can be significantly increased by inhibiting the uracil repair pathway.

We next investigate the toxicity of targeted deaminase in human cells. To test whether ZF-AID$^{\Delta NES}$ can be safely used as a genome editing tool without incurring DSBs, we generated a HEK293FT reporter cell line carrying a non-functional frame-shifted GFP, which could be rescued by DSB-enhanced HDR with an exogenous donor DNA[2,34] (GFP-In reporter in Fig. 4d). If DSBs occurred from ZF-AID$^{\Delta NES}$ treatment, GFP + cells can be detected. The GFP-In reporter also carries recognition sites for alternative nucleases, I-*Sce*I and ZF$_{GFPIN}$Ns (ZF$_{GFPINL}$N & ZF$_{GFPINR}$N)[34]. While expression of I-*Sce*I and ZF$_{GFPIN}$Ns generated 1.01 and 0.43% GFP + cells respectively, expression of ZF$_{GFPIN}$-AID$^{\Delta NES}$s gave rise to just 0.03% GFP + cells, which was further reduced to 0.01% if the UNG inhibitor UGI was co-transfected[32,35] (Fig. 4e). Thus, targeted deaminase with UGI treatment created 40-fold fewer DSBs at the target locus than alternative nuclease-based approaches, close to negative control levels where only the DNA donor was delivered. The extent to which ZF-AID$^{\Delta NES}$ generated DSBs (0.01%) was very low compared with its C:G→T:A editing activity (2.1%, Fig. 4b). Furthermore, we observed higher cell survival in the ZF$_{GFPIN}$-AID$^{\Delta NES}$s/UGI-expressing population (66%) than in the ZF$_{GFPIN}$Ns-expressing population (41%, Fig. 4f), suggesting that targeted deaminases are less cytotoxic than ZFNs. Thus, expression of chimeric AID$^{\Delta NES}$s with UGI enables efficient genome editing in human cells without generating DSBs and with low cytotoxicity.

## Discussion

We demonstrate that fusing cytidine deaminases with DNA binding modules enables site-specific deamination of genomic loci in both prokaryotic and eukaryotic cells. We designed and optimized the structure of programmable deaminases to effectively convert a specific C:G base pair to T:A in the *E.coli* genome, achieving up to 13% editing frequency. We then applied the optimized chimeric deaminases to a human cell line and found that these novel enzymes could create site-specific single-nucleotide transitions in 2.5% of cells. The transfected cells demonstrated decreased cytotoxicity compared with treatment with targeted nucleases.

A recent study independently developed targeted deaminases using Cas9 as the DNA binding domain and reported obtaining similar results, albeit by different means[16]. Their Cas9 deaminase achieved higher efficiencies by taking advantage of the fact that Cas9 binding generates an ssDNA loop, a natural substrate for AID and APOBEC deaminases. Instead of expressing UGI independently, they fused it to nickase-Cas9, and cleverly suppressed mismatch repair by allowing Cas9 to nick the non-targeted strand.

The authors of this study suggested that, due to their high efficiency and their avoidance of mutagenic dsDNA cuts, targeted deaminases might be promising tools for the correction of genetic diseases. While we agree that targeted deaminases are an excellent alternative tool for genome editing independent of DSB, our results suggest that further engineering is needed to reduce their processivity[16] and off-target activity. First, it is still difficult to pinpoint the activity of targeted deaminases to a specific cytidine within a ± 15 bp window (Fig. 3c,d), and we also detected off-target mutations >150 bp away from the DNA binding site. These findings indicate that targeting does not eliminate the processivity of deaminases due to enzyme sliding on substrate DNA. The processivity of deaminases has been reported to lead to local mutation clusters and chromosomal rearrangements (*aka*, kataegis and chromothripsis) and are associated with human breast and ovarian cancer[36,37]. Further engineering is needed to reduce enzyme sliding before them can be safely applied to therapeutic editing. Additionally, while the study from Komor *et al.* observed no apparent untargeted base editing surrounding

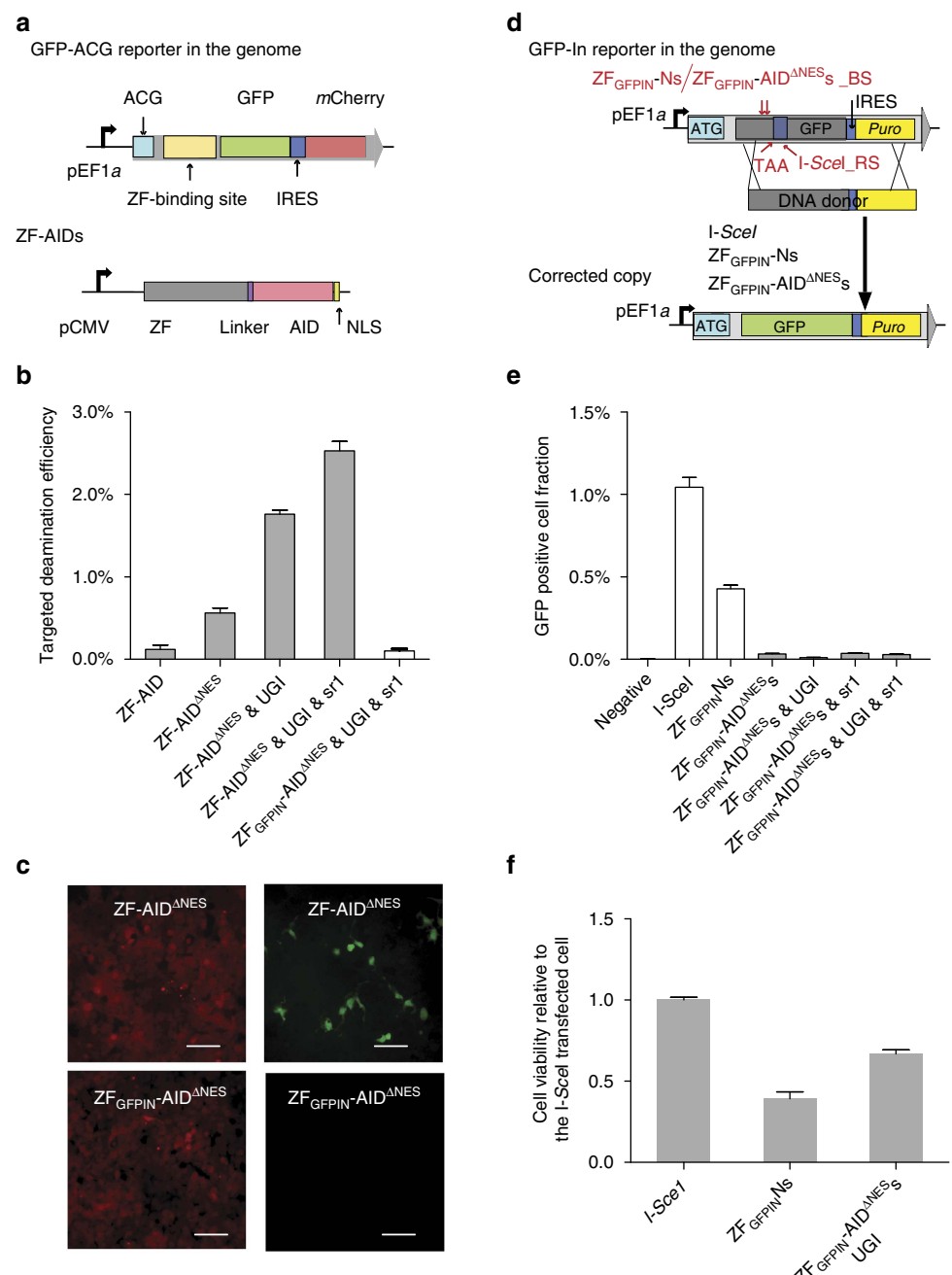

**Figure 4 | Targeted deamination and low toxicity of ZF-AID in human cells.** (**a**) Schematic representation of the ACG-GFP reporter system in HEK239FT cells (upper) and the ZF-AID (lower) tested for targeting deaminase activity. IRES, internal ribosome entry site; NLS, nuclear localization signal. (**b**) Targeted deamination activity of ZF-AIDs. ACG-GFP reporter cells were transfected with the constructs labelled on the X-axis. Targeted deamination frequency was estimated as the proportion of GFP-rescued cells 48 h after transfection. ZF-AID$^{\Delta NES}$ is identical to ZF-AID except with a deleted AID nuclear export signal (NES); UGI, inhibitor of UNG; sr1, shRNA-MSH2. (**c**) ACG-GFP reporter cells imaged under fluorescence (mCherry (left)/GFP (right)) 48 h after transfection with ZF-AID $^{\Delta NES}$/UGI/sr1 or ZF $_{GFPINL}$-AID$^{\Delta NES}$/UGI/sr1 plasmids. Scale bar, 200 μm. (**d**) Schematic design of DSBs assay. The genomically integrated GFP-In reporter includes a 35 bp frame-shift insertion bearing a stop codon and I-*Sce*I recognition site (I-*Sce*I_RS). Of note, ZF$_{GFPIN}$Ns and ZF$_{GFPIN}$-AID$^{\Delta NES}$s binding sites (ZF $_{GFPIN}$Ns/ZF$_{GFPIN}$-AID$^{\Delta NES}$s_BS) were identical and located 82 bp upstream of the insertion. We transfected the cells with a DNA donor carrying the wild-type GFP sequence along with I-Sce1/ZF$_{GFPIN}$Ns/ZF$_{GFPIN}$-AID$^{\Delta NES}$s expression plasmids and assessed the DSB-generating rate by measuring HDR frequency as determined by GFP rescue of the cells. (**e**) GFP rescue results determined by flow cytometry. Negative control was transfected with the DNA donor only. (**f**) Cytotoxicity assay for ZF$_{GFPIN}$-AID/UGI relative to I-Sce1. A value of <1 shows decreased cell survival as compared with I-*Sce*I, and demonstrates a toxic effect.

on-target and off-targete Cas9 loci, our whole genome sequencing data demonstrated elevated levels of global deamination in off-target WRC sites (Fig. 3e,f), suggesting that deaminases maintain their intrinsic DNA binding preferences and editing activity even when fused to targeted DNA binding domains. Most APOBEC proteins except for APOBEC3B are carefully kept out of the nucleus via strong nuclear exportation signals[38]. The overexpression of APOBEC3B has been implicated as a driver for

human breast, ovarian and cervical cancers through its generation of random C:G→T:A transitions[39,40], which provides a cautionary note about the need to constrain the intrinsic preferences and activity of deaminases. One possible solution to this problem would be to create obligatory dimeric targeted deaminases by splitting the deaminase protein and fusing each half to an independent DNA binding domain, so that an active deaminase protein would only be generated if both halves are targeted to two specific nearby DNA sequences. This approach could also reduce the processivity of the enzyme.

Our results set the stage for the future engineering of additional targeted DNA nucleotide mutases[41] beyond cytidine deaminases, such as targeted adenosine deaminases[42], that effect changes to DNA without introducing DNA cuts or nicks. In this study, we did not observe targeted genome editing activities of other deaminases (APOBEC1, 3F, 3G, Fig. 1a), however, we did not exclude the possibility that expression level, as well as sequence preference contributed to the negative signals. We look forward to future studies to exploring other customized deaminases for different target sequences. Aside from using such targeted mutases for gene therapy as suggested for deaminase in the recent study[16], suitably engineered to reduce their intrinsic mutability as we indicate, we foresee other potential uses of these tools. First, as nucleases tend to be toxic in prokaryotes[43,44] due to their lack of efficient non-homologous end joining pathway, targeted DNA mutases may provide an effective means to engineer prokaryotic species such as Streptococcus for which few molecular tools are available. As demonstrated here with deaminases, targeted mutases have potential to be highly portable in both prokaryotic and eukaryotic systems. Second, although DSBs are better tolerated in eukaryotes, certain cell types are very sensitive to DSBs incurred by nucleases, such human induced pluripotent cells. We demonstrated that targeted deaminases incur significantly less cytotoxicity compared with nucleases in HEK293 cell and we envision that along with other targeted mutases, they will similarly be less toxic in these sensitive cell types. Moreover, the independency from exogenous DNA donor to make precise mutations likely allows this tool to be efficient in the agriculture applications without GMO regulation, similar to recently successful cases of mushroom and corn in which CRISPR was used to disrupt endogenous genes. In addition, targeted mutases should be effective ways to generate precise mutations in non-dividing cells in which HDR activity is extremely low[13]. One of the limitations of homology directed repair is that it is restricted to the G2/S phase and thus barely applicable to cells with a dormant cell cycle. Targeted deaminases can potentially bypass such a problem by introducing C-to-U mutations on the template strand of dsDNA, which can be recognized by RNA polymerase II to produce RNA with intended mutations independent of the cell cycle. Finally, although our group demonstrated highly multiplexible (62X) targeting of a repetitive sequence in immortalized pig cells[45], it has proven highly difficult to achieve this result in primary cells, likely because the high number of DSBs required to achieve this result leads to chromosomal rearrangements, senescence and apoptosis. We believe that the independence from DSBs may make targeted mutases a safer and more efficient tool for editing and studying repetitive elements in the genome.

## Methods

### Construction of fusion proteins.
To construct ZFP-AID fusion proteins, we first PCR amplified ZFP from pUC57-ZFP (ref. 18) and AID from pTrc99A-AID (ref. 46) and fused these two parts with various linkers using overlap PCR. The fusion constructs were cloned into a pTrc-Kan plasmid. We fused AID with TALE by cloning AID into pLenti-EF1a-TALE(0.5 NI)-WPRE (ref. 20) plasmid and then cloned TALE-AID fusions into the pTrc-Kan plasmid. APOBEC1, 3F and 3G genes were synthesized (Genescript) and cloned into the pTrc-ZFP-Kan plasmid.

To generate pCMV-ZF-AID constructs, we amplified the ZF-AID cassette from pTrc-ZF-AID and cloned it into a pCMV-hygo[20] plasmid. The detailed construction methods are illustrated in Supplementary Fig. 1. The sequences of the fusion proteins are listed in Supplementary Note 1–5.

### Construction of *E.coli* reporter cell lines.
The GFP coding sequence was amplified from pRSET-EmGFP (Invitrogen). We modified the reporter by mutating the start codon to ACG and inserting a ZFP/TAL binding site upstream of the GFP coding sequence. To establish stable cell lines with a single copy of the GFP reporter sequence in the genome, we integrated the GFP cassette into the galK locus in the EcNR1 (MG1566 with λ-prophage::bioA/bioB) and EcNR2 (EcNR1 with mutS knocked out) strains[19]. To knock out ung, we replaced the ung gene with Zeocin resistance cassette via recombineering. In addition, all the reporter cell lines were transformed with pTac-T7polymerase to induce the expression of GFP. Subsequent modifications of the reporter were conducted using the MAGE system[19]. All sequences can be found in Supplementary Note 6 and 7.

### E.coli cell culture and targeted deaminase activity assay.
The reporter strains were electro-transformed with the plasmids coding for targeted deaminases. Single colonies were inoculated and cultured under 34 °C in LB-min- media (5 g NaCl, 5 g yeast extract,10 g tryptone in 1L ddH$_2$O) supplemented with 100 μg ml$^{-1}$ Carbenicillin, 25 μg ml$^{-1}$ Chloramphenicol, 100 μg ml$^{-1}$ Spectinomycin and 100 ug ml$^{-1}$ Kanamycin. Targeted deaminase activities of the targeted deaminases were tested by inducing the expression the fusion protein with IPTG of final concentration 100 μM when the O.D. of the cell culture reached 0.4–0.6. To maintain the continuous cell proliferation, cell culture was diluted 100-fold into fresh media every 10 h.

### Flow cytometry.
Targeted deaminase activity as measured by GFP + cell fraction in the total population was assayed by flow cytometry using a LSRFortessa cell analyzer (BD Biosciences). Bacteria culture was diluted 1:100 with PBS and vortexed for 30 s before flow cytometry. At least 100,000 events were analysed for each sample. Targeted deamination efficiency was calculated as the percentage of GFP positive cells in the whole population.

### GFP gene Sanger sequencing.
To genotype the GFP and GAPDH genes in E.coli, we inoculated single colonies in LB media and cultured them for 16 h at 34 °C. PCR reactions with Phusion enzyme (NEB) were conducted with 1 μl 100 × diluted bacterial culture and Sanger sequencing were performed. The sequence of primers can be found in Supplementary Note 11.

### Genomic library preparation.
Corresponding reporter strains were transformed with ZFP-8aa-AID and TALE-C3-AID, respectively. Single colonies were inoculated and split into the induction and non-induction groups. The expression of the deaminases was induced for 10 h and the cell culture was plated on IPTG containing agar plate to isolate single colonies. After ∼24 h, we inoculated single colonies in LB-min media and cultured them overnight at 34 °C. To extract chromosomal DNA and minimize the amount of plasmid DNA, Miniprep was first performed (Qiagen) according to manufacturer's protocol and the sodium acetate/SDS precipitate formed was resuspended in the Lysis Buffer Type 2 (Illustra bacteria genomicPrep Mini Spin Kit, GE Healthcare), and the genomic DNA was recovered following manufacturer's instructions (Illustra bacteria genomicPrep Mini Spin Kit, GE Healthcare). Genomic DNA libraries were constructed from 1.5 to 2 μg of genomic DNA. DNA was sheared in TE buffer (10 mM Tris (pH 8.0) 0.1 mM EDTA) using microTube (Covaris) with recommended protocol. Median DNA fragment sizes, estimated by gel-electrophoresis, were 150–250 bp. Sheared fragments were processed with the DNA Sample Prep Master Mix Set 1 (NEB). Adaptors consisted of the Illumina genomic DNA adapter oligonucleotide sequences with the addition of 2-bp barcodes. Eight barcoded genomic libraries were pooled with equal molar amount. The sequences of the adaptors and primers can be found in Supplementary Note 12.

### Genomic DNA sequencing analysis.
The reference genomic sequence for the reporter strain was generated by manually modifying the FASTA sequence of E. coli K-12 strain MG1655 to reflect the removal of mutS and ung, the insertion of the lambda prophage genome into the bioAB operon, and the insertion of the GFP reporter into the galK cassette. Genomic libraries were single-end sequenced using an Illumina Genome Analyzer, generating 100 bp reads. The reads were first assigned to samples according to their 2-bp barcodes by exact matching and reads with fewer than 60 bases of high-quality sequence were discarded. Sorted reads were then aligned to the reference genomes using the Breseq package. Match lengths of at least 40 bases were required for alignment. In addition to Breseq's single nucleotide variation (SNV) calling functionality, the SAMtools package[47] was used on the resulting BAM file to corroborate short indels and single nucleotide variants. To validate the result of Breseq, MAQ was used as a second method to align the raw reads to the reference genomes and to call SNVs. FastQ files containing the sequencing reads were split based on the barcode, and trimmed

using the FASTX-toolkit library. The resulting fastQ files were mapped to the reference genomes with MAQ (ref. 48). Single nucleotide substitutions were considered valid when supported by a minimum read depth of 10 or a Phred-like consensus quality higher than 80. Finally, these three sets were merged to generate the final SNV set. SNVs called by both MAQ and SAMtools, or SNVs called by one and also called by Breseq, were kept. Indels were called by SAMtools alone. Breseq was used to identify new junctions using candidates generated by split read alignment. LiftOver[49] was used to map the SNVs back to the original MG1655 genome (NCBI accession: NC_000913) for annotation. SNV effect prediction was done using the snpEff package[50] and BioPerl. The analysis flow map (Supplementary Fig. 5) provides information about the number of raw sequence reads, aligned reads, genome coverage and validated SNVs (Supplementary Tables 1 and 2), and the list of SNVs (Supplementary Data 1) can be found in supplementary information.

**Statistical analysis of the whole-genome sequence data.** Wilcoxon test was used to analyse whether the mutation rate was higher in the strains with TALE-AID or ZF-AID induction. Intended mutations in TALE-AID, and ZF-AID strains were discarded for this analysis. $X$ (SNVs not induced) = 31, 21; $Y$(SNVs induced) = 40, 31, 27, 33, 20, 32. $H_0$: There is no difference in the mutation rate; $H_1$: induced strains have a higher mutation rate. $P_{value} = 0.25$. The null hypothesis cannot be rejected. Therefore, there was no significant difference in the number of mutations between induced and uninduced strains. Due to the limited sample size, sensitivity simulations were performed to ensure an appropriate type II error. (1) Random samples from the observations of size m + n were taken, and divided in two groups A (n members) and B (m members). (2) An arbitrary value δ was added to B. (3) Wilcoxon one-sided $P$ value was calculated for comparison of groups A and B. A $P$ value under 0.05 was considered a success and recorded; otherwise a failure was recorded. (4) Steps 1 to 3 were repeated 10,000 times. The estimated power of the test was approximated by the proportion of successes among the 10,000 repetitions. (5) Steps 1 to 4 were repeated for a range of values of m + n and a range of δ values. The results are presented in Supplementary Fig. 6. With the current sample size, we could detect an increase of 13 SNVs, or higher (Statistical power = 0.8).

**Poisson based modelling of number of genome edited sites.** There are four sites in the genome with equivalent features as the targeted site. All of them contain an exact ZF binding sequence 11 bp away from an upstream WRC motif. Deamination was only detected in the targeted site with a maximum frequency of 7%. Assuming that alterations of these sites are Poisson distributed with $\mu = 0.07$, the probability of detecting a second mutation in any strain is 0.03, and the probability $P$ of *not detecting* an additional mutation in any of the 3 ZF-AID strains is 0.90.

$$P(k > 1 \mid k \geq 1) = \frac{1 - P(k = 0) - P(k = 1)}{1 - P(k = 0)} = \frac{1 - e^{-0.07} - 0.07e^{-0.07}}{1 - e^{-0.07}} = 0.03$$

$$P = (1 - P(k > 1 \mid k \geq 1))^3 = 0.90$$

**Human cell culture.** The human embryonic kidney cell line HEK293FT (Invitrogen) and the derivative reporter cell lines was maintained under 37 °C, 5% CO$_2$ using Dulbecco's modified Eagle's Medium supplemented with 10% FBS, 2 mM GlutaMAX (Invitrogen), 100 U ml$^{-1}$ penicillin and 100 µg ml$^{-1}$ streptomycin.

**Targeted deaminase activity assay.** The GFP-ACG reporter cell lines were generated by lentiviral transduction with low virus titration to make sure at most one copy of the reporter can be integrated into the genome. Single colonies were isolated via FACS based on mCherry signal (Beckman Coulter MoFlo). Deaminase activity was tested by transfecting reporter cells with plasmids carrying ZF-AIDs. Briefly, HEK293FT cells were seeded into 12-well plates the day before transfection at densities of $4 \times 10^5$ cells per well. Approximately 24 h after initial seeding, cells were transfected using Lipofectamine 2000 (Invitrogen) and 1.6 ug DNA (400 ng of ZF-AID expression plasmid and/or 20 ng of UGI expression plasmid, and/or 20 ng of ShRNA-MSH6 expression plasmid (Sigma), and pUC19(Invitrogen) plasmid to 1.6 ug) per well. After 48 h, cells were trypsinized from their culturing plates and resuspended in 200 µl of media for flow cytometry analysis. At least 25,000 events were analysed for each transfection sample. The flow cytometry data were analysed using BD FACSDiva (BD Biosciences). The reporter and constructs sequences can be found in the Supplementary Notes 13–15.

**Genotyping of human cell.** To genotype the GFP target locus in HEK293 cells, we picked single GFP + monocolonies and added each to 10 ul 1 × prepGEM buffer and enzyme (ZyGEM). After cell lysis according to the manufacturer's instructions, the bulk product was added to a PCR reaction containing Platinum Taq polymerase (invitrogen). PCR products were cloned in pCR4-TOPO (invitrogen) and capillary sequenced by Genewiz.

**DSB generating potential assay.** The GFP-In reporter[34] cell lines were generated by lentiviral transduction and successful reporter insertions were selected via puromycin selection. GFP-In reporter cells were plated in 12-well plates the day before transfection at densities of $4 \times 105$ cells per well transfected using Lipofectamine 2000 (Invitrogen) and 1.6 µg DNA (400 ng of ZF-AID/ZF-nuclease/I-Sce1 expression plasmid and/or 20 ng of UGI expression plasmid, and/or 20 ng of ShRNA-MSH6 expression plasmid, 1 µg of DNA donor pUC19 (Invitrogen) plasmid to 1.6 ug) per well. After 72 h of transfection, cell were trypsinized and resuspended in 200 µl of media for flow cytometry analysis. At least 25,000 events were analysed for each transfection sample. The flow cytometry data were analysed using BD FACSDiva (BD Biosciences). The constructs sequence can be found in the Supplementary Note 16.

**Cytotoxicity assay.** The assays were conducted as previously described[51]. Briefly, HEK293FT cells were seeded in 12-well plates ($4 \times 10^5$ cells per well) and transfected after 24 h with 200 ng of deaminase/nuclease expression plasmids, 10 ng of pmaxGFP (Lonzon), and pUC19 to 2 µg using calcium phosphate-mediated protocol. After 2 and 5 days, the fractions of GFP-positive cells were determined by flow cytometry (BD Biosciences). Survivability was calculated as the percentage of GFP-positive cell surviving at day 5 divided by the percentage of GFP-positive cells determined at day 2 after transfection. This ratio was normalized to the corresponding ratio after I-SceI transfection, to yield the percentage survival as compared with I-SceI.

**MAGE and dsDNA-mediated homologous recombination.** Single colonies were inoculated into LB-min media and cultured under 34 °C to an absorbance (600 nm) of 0.4–0.6. The bacterial culture was then shifted to 42 °C for 15 min to induce expression of the λ-Red recombination proteins (Exo, Beta and Gam), and then immediately chilled on ice (up to 2 h). 1 ml of bacterial culture was centrifuged at 16,000g for 30 s and washed twice with 1 ml dH20 at 4 °C. Cell pellets were re-suspend with 50 µl DNA-containing water (100 ng dsDNA fragment or 200 pmole ssDNA) and transferred to a pre-chilled 1 mm gap electroporation cuvette (Bio-Rad), and electroporated with a Bio-Rad GenePulser electroporation system under the following parameters: 1.8 kV, 200 Ω and 25 µF. 1 ml S.O.C (New England Biolabs) was immediately added to the electroporated cells. The cells were recovered in S.O.C at 34 °C for 2–2.5 h before plating on LB-min agar plates to resolve single colonies. The plates were incubated for at least 13 h at 34 °C. Colony PCR followed by Sanger sequencing was performed to screen for colonies with the right genotypes.

**Genotyping the GFP and GAPDH loci.** To genotype the GFP and GAPDH genes, we inoculated single colonies in LB media and cultured them for 16 h at 34 °C. gfp and gapdh loci were amplified from 1 ul of 100 times diluted bacterial culture using 10 µl 2 × Phusion High-Fidelity PCR Master Mix (NEB), 7 µl water, and 1 µl of 10 µM primer(each) with thermocycling program of 98 °C for 2 min; (98 °C for 30 s, 60 °C for 30 s, 72 °C for 2 min) × 30 cycles, 72 °C for 10 min. The primer sequences can be found in Supplementary Note 11.

**Preparation of whole-genome sequence library.** As illustrated in Supplementary Fig. 7, the reporter strain for testing ZF-AID-targeted deamination activity was transformed with ZF-8aa-AID and one single colony was inoculated. Cells from the colony were cultured in 2 ml LB-min media supplemented with 100 µM IPTG to induce the expression ZF-8aa-AID. In parallel, cells from the same colony were cultured in 2 ml LB-min media without induction. After 10 h, the bacterial culture was plated on the IPTG-containing agar plate and 3 GFP + colonies (after IPTG induction) and 1 GFP − colony (negative control without IPTG induction) were inoculated and cultured in 2 ml LB-min media overnight at 34 °C. The same work flow was undertaken for experiments with TALE-C1-AID and its corresponding reporter strain.

To extract chromosomal DNA and minimize the amount of plasmid DNA, Miniprep was first performed (Qiagen) on the 2 ml bacteria culture according to manufacturer's protocol. The sodium acetate/SDS precipitate formed was resuspended in the Lysis Buffer Type 2 (Illustra bacteria genomicPrep Mini Spin Kit, GE Healthcare) and the genomic DNA was recovered following manufacturer's instructions. Genomic DNA libraries were constructed from 1.5 to 2 µg of genomic DNA. DNA was sheared in TE buffer (10 mM Tris (pH 8.0), 0.1 mM EDTA) using microTube (Covaris) with duty cycle as 10%, intensity as 5, cycle per burst as 200 and time as 780 s per sample. Median DNA fragment sizes as estimated by gel-electrophoresis, were 150–250 bp. Sheared fragments were processed with the DNA Sample Prep Master Mix Set 1 (NEB). Adaptors consisted of the Illumina genomic DNA adapter oligonucleotide sequences with the addition of 2-bp barcodes. Eight barcoded genomic libraries were pooled with an equal molar ratio. The sequences of the adaptors and primers can be found in Supplementary Note 12.

**Plasmid and E.coli strain construction.** Construction of targeted deaminase expression vectors: Restriction enzymes and Rapid ligase were purchased from New England Biolabs and used according to the manufacturer's instructions. PCRs were conducted by Kapa HiFi PCR 2× master mixture (Kapa Biosystems).

The primers and oligos were obtained from IDT and gene synthesis was provided by Genescript. The schematic of various constructions tested in this study is in Supplementary Fig. 1. All of the primer, construct and backbone sequences are listed in Supplementary Notes.

*Construction of pTrc-Kan as the backbone vector.* We first constructed a common inducible expression vector by combining the elements from pZE-21 and pROEX-HTa vectors. In brief, the fragment containing lacI gene and pTrc promoter of pROEX-HTa was amplified by PCR using lacI-XhoI and pTrc-HindIII primers. This fragment was digested with XhoI and HindIII and ligated into a similarly digested pZE-21 to make pTrc-Kan. A NheI restriction site was also imbedded downstream of the pTrc promoter for future cloning.

*Construction of pTrc-ZF and pTrc-AID.* The ZF gene[18] was amplified from pUC58-ZFP by PCR using ZF-F and ZF-R-HindIII primers. ZF fragments were digested with HindIII and NheI and ligated into the pTrc-Kan backbone plasmid, which was similarly digested. The AID gene was amplified from pTrc99A-AID, a gift from Meng Wang[46], using primers AID-F-NheI and AID-R. AID fragments were digested with HindIII and NheI and ligated into the pTrc-Kan backbone plasmid digested with the same enzyme.

*Construction of pTrc-ZF-AIDs.* The ZF was appended to the N-terminus of AID using amino acid linkers of various sizes and composition. The ZFP gene was amplified from pZFPerb2 and the linker sequence was created by PCR using ZFP-F and ZFP-R (4aa, 4aa2, 8aa and 11aa) primers individually. In parallel, the AID gene was amplified from ptrc99A-AID and the linker sequence was created by PCR using AID-F (4aa, 4aa2, 8aa and 11aa) and AID-R primers. ZF and AID with corresponding linkers were fused by overlap extension PCR using ZF-F and AID-R primers. Each construct was digested with NheI and HindIII and ligated into the four similarly digested pTrc-Kan backbone plasmids.

*Construction of pTrc-ZF-APOBECs.* We constructed pTrc-ZF-APOBECs with various linkers using the isothermal assembly protocol[52]. In brief, the APOBEC1, APOBEC3F, APOBEC3G 2K3A genes were amplified by PCR using primers APOBEC-F and APOBEC-R. pTrc-ZFP-aaAID was linearized by SalI and HindIII digestion and the pTrc-ZF fragment was recovered by gel purification. The pTrc-ZF fragment was fused to individual APOBEC fragments by isothermal assembly.

*Construction of pTrc-TALE-AIDs.* AID gene was amplified from pTrc99A-AID plasmids, digested with NheI and BsrG1 and cloned into the pLenti-EF1a-TALE(0.5 NI)-WPRE 20, which was similarly digested. The obtained TALE-C1-AID fusion was amplified and digested with SspI and HindIII and inserted into the pTrc-Kan backbone vector to obtain the pTrc-TALE-C1-AID construct. TALE truncations were created by amplifying the TALE fragment with the appropriate TALE-F and TALE-R deletion primers. The truncated TALEs were then ligated into pTrc-TALE-AID plasmid using the SspI and NheI sites.

*Construction of pTrc-TALE-APOBECs.* APOBEC genes were each amplified by PCR using APOBEC-F-NheI and APOBEC-R-HindIII primers individually, digested with NheI and HindIII, and ligated into the pTrc-TALE-AID plasmid that was similarly digested.

*Construction of pL-tetO-ZF-AIDs.* The pL-tetO promoter was amplified from the pZE-21G plasmid by PCR using primers pL-tetO-5 and pL-tetO-3. This fragment was digested with NheI and XhoI and ligated into pTrc-ZF-AID plasmid that was similarly digested.

*Construction of pCMV-ZF-AIDs.* pCMV-ZF-AID/pCMV-ZF-AIDΔNES constructs were built by amplifying ZF-AID/ZF- AIDΔNES using BsiWI -ZF and BsrGI-AID/BsrGI-ΔAID primers, respectively. The PCR products were digested with BsrGI and BsiWI, and ligated into pCMV-puro backbone that was similarly digested. To generate ZFGFPIN-AIDΔNES expression vectors, ZFGFPINL and ZFGFPINR were amplified from pST1374-G223L and pST1374-G223R (ref. 53) vectors using BsiWI-ZFL/R, BamHI-ZFL/ZFR primers and cloned into pCMV-ZF-ΔAID to swap the ZF domains using BsiWI and BamHI. Subsequently, ZFGFPIN-AIDΔNES were amplified and cloned into pST1374 vector by NheI and ApaI restriction sits to generate ZF*-AIDs expression vectors with the same backbone and DNA binding module as ZFGFPIN-Ns (ZFGFPINL-Ns/ZFGFPINR-Ns).

*Construction of pCMV-UGI.* UGI encoding gene optimized for human cell expression was synthesized and cloned into pCMV-puro using XhoI and BsiWI restriction sites. The sequence and primers can be found in Supplementary Note 14.

*Construction of bacterial reporter systems.* We conducted all optimization experiments in the EcNR1 (MG1566 with λ-prophage::bioA/bioB) and EcNR2 (MG1566 with λ-prophage::bioA/bioB and tetR/cmR::mutS)[19]. *E. coli* strains. To build a single copy GFP reporter in the genome, we integrated a modified GFP cassette into the bacterial genome through λ-red recombineering[19]. Additional modifications on the GFP reporter were achieved through the Multiplex Automated Genome Engineering (MAGE) system developed in the lab[19]. The *ung*-knockout was achieved through dsDNA-mediated homologous recombination.

*Construction of the GFP reporter strains.* First, we modified the GFP cassette on pREST-EmGFP (Invitrogen) to contain a ZFP binding site. A dsDNA fragment was synthesized with the ZFP binding site (5′ GCCGCAGTG 3′) 9 bp downstream of a start codon, and the fragment was flanked by the NdeI and NheI restriction sites. This fragment was digested with NdeI and NheI and ligated to a similarly digested pREST-EmGFP to construct pREST-ZFP-EmGFP. Modified GFP cassette was incorporated into the galK locus in the EcNR1 and EcNR2 strains using the λ-red

recombineering. In brief, the GFP cassette was amplified by PCR using 5′-galk-gfp and 3′-gfp-galk primers to create galK homology on both sides of GFP. This fragment was transformed into λ-red-induced strains, and successful insertions were selected for based on GalK negative selection[54]. Subsequently, we modified the single-copy GFP reporter through MAGE. To control the expression of GFP from the T7 promoter, we introduced the plasmid pTac-T7 RNA polymerase, in which T7 RNA polymerase is transcribed from tac promoter of the lactose operon.

*Construction of the ΔmutS Δung strain.* We used the EcNR1 strain as the *mutS+ ung+* background and the EcNR2 strain as the ΔmutS background to test the targeted deamination frequency. To obtain the ΔmutS Δung background, we disrupted the ung gene in the EcNR2 strain by inserting a Zeocin resistance cassette in the middle of the gene. In brief, a Zeocin resistance cassette flanked by ung homology regions was PCR amplified from pEM7-Zeo vector (Invitrogen) using the 5′ung-zeo and 3′zeo-ung primers. This PCR fragment was transformed into the EcNR2 reporter strain. Successful disruption of ung was selected based on Zeocin resistance.

**Data availability.** The authors declare that all the data supporting the findings of this study are available within the paper and its supplementary information files. The whole genome sequencing data has been uploaded to Sequence Read Archive under the accession code SUB1973749.

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

## Acknowledgements

We acknowledge current and previous members of the Church Lab for their helpful discussion and support.

## Author contributions

L.Y. and G.C. conceived the ideal. L.Y., G.C., J.A., F.Z. and P.M. designed the experiment. L.Y. and W.L.C. conducted all the experiments with the assistance of D.C. and V.S., L.Y., M.G., D.B.G. and J.A. performed the data analysis. L.Y. and W.L.C. wrote the manuscript with the help of A.W.B., J.A., F.Z., E.L. and other co-authors.

## Additional information

**Competing financial interests:** L.Y. and G.C. filed the patent related to this manuscript with the application number: US 12/939,505. The remaining authors declare no competing financial interests.

DOI: 10.1038/ncomms16169    OPEN

# Corrigendum: Engineering and optimising deaminase fusions for genome editing

Luhan Yang, Adrian W. Briggs, Wei Leong Chew, Prashant Mali, Marc Guell, John Aach, Daniel Bryan Goodman, David Cox, Yinan Kan, Emal Lesha, Venkataramanan Soundararajan, Feng Zhang & George Church

Nature Communications 7:13330 doi: 10.1038/ncomms13330 (2016); Published 2 Nov 2016; Updated 9 Oct 2017.

The original version of this Article contained an error in the email address of the corresponding author George Church. The correct email is gchurch@genetics.med.harvard.edu. The error has been corrected in the HTML and PDF versions of the Article.

