## [Peer Review File · Nature Communications]

Reviewers' comments:

Reviewer #1 (Remarks to the Author):

In the manuscript entitled "genome editing with targeted deaminases", Church and colleagues examine the ability of mammalian DNA deaminases to be targeted to defined loci and induce genome editing. The authors express chimeric zinc-finger-DNA deaminase variants in bacteria and test the ability of the deaminases to revert an ACG to an ATG codon in GFP. They demonstrate that targeted deamination is most robust with AID and that the deamination frequency is specific to the target sequence, and there are very few non-specific mutations. The novel chimeric proteins were also demonstrated to function in human cells on chromosomally integrated substrates without significant associated toxicity. These experiments led the authors to propose that targeted DNA deamination by AID and related deaminases could be a convenient way to generate targeted mutations, especially in systems where DNA cleavages are toxic.

With the advent of CRISPR/Cas9, much effort has been invested of late in designing strategies for robust targeted DNA manipulation. Even the APOBEC/AID family members have been demonstrated to function in precise genome editing by David Liu and colleagues. But the field continues to evolve with more efficient targeting strategies. Thus, new techniques, especially those with fewer collateral damage, as described in this manuscript, are of much interest. However, there are several issues with this manuscript that should be resolved before publication.

1. The authors should include catalytically inactive AID in their reversion assays to be confident that the effect they see is indeed DNA deamination by AID. Several catalytic mutants of AID have been described by others. A minor point, the catalytic domain is not in the C-terminus of AID as the authors have indicated.
2. An important control that is missing throughout the manuscript is expression levels of the various chimeric molecules. There are no western blots for AID expression, and without this data it is impossible to be convinced that the differences in deamination activity is really due to functional modulation of the deaminase and/or targeting sequence.
3. It would be important to test if the deamination target matches that of the deaminase used. AID and APOBEC3 for example have different target specificities. Can the ACG sequence used in the reversion assay be altered so that now APOBEC3G is a better deaminase for editing? This kind of data is important as it would allow one to customize deaminases for different target sequences.
4. The results generated with the C-terminal truncation mutation is interesting but it is important that this experiment is better controlled. The C-terminal deletion dramatically destabilizes AID, even though there is increased frequency of switch region mutation in B cells, and it is only under specific cases such as an ER-fused AID wherein the truncated protein is stabilized. Overall, this is a very complicated mutation. Thus, it is important to characterize at least the expression level of the protein in the context of the assay.

Reviewer #2 (Remarks to the Author):

The paper by Yang and coworkers examines cytidine deaminases conjugated with zinc finger (ZF) proteins or transcription activator-like effectors (TALE) as a new genome editing enzyme unit. The

authors tested candidate deaminases for genome editing activity using ZF proteins or TALEs as the DNA binding module and optimized the conditions for better editing efficiency using reporter systems in *E. coli*. The authors then identified the off-target activity pattern of the deaminases and found increased global C-to-T background mutations. They also observed that these novel enzymes also function in human cells at an efficiency of 2.5%.

A similar but more comprehensive and potentially more significant study using Cas9 nickase combined with cytidine deaminase was published by David Liu's group in *Nature* April 20, 2016 [Komor et al. (ref. 30)]. The study under review is different from the Liu group's study in that the authors used ZF proteins or TALEs as the DNA binding module. However, both the broad range of C-to-T conversion near the binding sites and the low efficiency of the current enzyme limit the significance and potential applications of this enzyme for precise genome editing.

Major comments:

1. The low deaminase efficiency shown here clearly limits the significance of this study. The authors should seek more ways to improve the efficiency of targeted deaminases. Possible approaches may include, but are not restricted to, the following:

1.1. Has the binding of the relevant ZF proteins or TALEs to the target DNA been validated? The low deaminase activity might be due to improper binding of the enzyme to the target DNA sequence, which the authors should check.

1.2. The authors of the current study tethered ZFs or TALEs to the N-termini of various deaminases and found that AID showed comparatively robust efficiency among those tested. The recent work by Komor et al. (ref. 30) examined deaminase activity in mammalian cells after tethering Cas9 to deaminase C-termini and found that rAPOBEC1 was the better genome editing deaminase. These different results might be caused by the different DNA binding units, ZF or TALE versus Cas9, used in the two studies and/or fusion of the DNA binding units to the N- versus C-termini. The authors could check whether changing either of these variables would enhance efficiency in their system.

1.3. Komor et al. (ref. 30) fused uracil DNA glycosylase inhibitor (UGI) to the C-terminus of Cas9 to increase efficiency, a possibility that should be tested in this study as well. Using mutant bacterial cell lines or perturbing cell behavior with a UGI inhibitor and MSH2 shRNA, as proposed by the authors, could significantly limit the application of the methods.

2. In the 3rd paragraph of the section titled 'Specificity of targeted deaminases', the authors indicated that the activity window is +/- 15 bp. Actually, the chance of only one C occurring within 30 bp is very rare and this method would be difficult to apply for precise genome editing.

3. The authors did not mention the distance between the ZF binding site and the target site in the main manuscript, although supplemental figure 1a and supplemental method 6 describe the distance as 9 bp. Because the distance could be critical for enzyme activity, it would be beneficial if the authors could explain the distance determination in a greater detail.

4. After determining the importance of linker length using ZF-AID, the authors optimized TALE-AID activity by truncation of the TALE C-terminus, not by regulating linker length. This decision requires an explanation.

5. Figure 3c: ZF-AID showed off-target effects at sites >150bp away from its original target site. The 9 bp ZF binding sequence is somewhat short because such a site might occur more than once in a genome. If the authors would analyse the sequences around these unwanted mutations and in addition use longer ZF binding sequences, they might find the reason for these off-target effects.

6. Figure 3c-f: The authors sequenced bacterial genomes to assess genome-wide off-target effects; however, they used only one uninduced control per experiment. Considering the variation between the

two uninduced controls in e and f, we suggest testing at least two or three uninduced controls per experiment to allow more solid conclusions to be reached about genome-wide off-target effects.

7. There are quite a few typos and errors. The following are some of the examples.

7.1. In the [Specificity of targeted deaminases] section: Typographical errors in the sentence that starts with "In the GFP- population, the only variant..."

7.2. Figure 1b: "deamianse" should be changed to "deaminase".

7.3. Figure 2c: "NLS" should be changed to "Linker".

7.4. Supplemental figure 1a: Start codons of dimers should be corrected to ACG.

Minor comments:

1) The species of the tested deaminases should be mentioned.

2) Some of the figure citations in the text do not match with what is shown in the figures. I would suggesting checking every figure citation.

Example: In the [Optimization of targeted deaminases] section, line 23: Fig 2e is cited, yet Fig 2 does not contain a section "e".

3) In the [Optimization of targeted deaminases] section, line 3: "a longer linker length improved editing frequencies" is an ambiguous explanation of the data. An enzyme containing an 11-amino acid linker showed lower activity than one with an 8-amino acid linker.

4) In the [Optimization of targeted deaminases] section, lines 10-11: "C-terminus of TALE" would be more accurate than "C-terminus"

5) In the [Specificity of targeted deaminases] section: The authors claimed that TALE-C3-AID showed strong sequence specificity for the first 8 bp. When I counted the bases in this region in the figure, it is 9 bp rather than 8 bp. The authors should check the number again.

6) Introduction: One of the limitations of homology directed repair is that it is restricted to G2/S phase and thus barely applicable to cells with a dormant cell cycle. Adding this point will strengthen the manuscript.

Dear reviewer,

Thank you so much for your valuable comments! We really appreciate the insightfulness of your comments, and we have made substantial revisions to incorporate them into our manuscript.

First of all, we would like to emphasize that the objective of our study is to characterize the tool of targeted deaminases, and we observed potential off-target deamination due to the intrinsic deaminase activities.

A recent publication (Komor, 2016, Nature) achieved up to 75% editing efficiency using a Cas9 deaminase, which elicits an enormous amount of interest in applying this tool in correction based gene therapy. In our study, we have independently developed this tool and performed similar optimizations, which was published in a patent in 2009 and a thesis in 2013. Compared to Komor et al, we were not able to push this technology to its limit since some of the tools were not available to us when the bulk of this study was conducted. However, we realized two major drawbacks in its application in therapeutic editing:

1. Targeted deaminases demonstrated pronounced processivity due to the enzyme sliding on single-stranded DNA substrates.
2. Targeted deaminases significantly increase the global deamination due to off-target binding at the deaminase recognition sites.

These observations raised concerns of tumorigenesis for gene therapy considering the biology of the APOBEC deaminases, and we proposed potential strategies to warrant specificity in their therapeutic applications. Since pronounced off-target effects could be already observed using our current targeted deaminases, we believe further optimizations on their activity are unlikely to strengthen our conclusions.

Below find a point-by-point response to your comments:

Reviewer #1 (Remarks to the Author):

In the manuscript entitled "genome editing with targeted deaminases", Church and colleagues examine the ability of mammalian DNA deaminases to be targeted to defined loci and induce genome editing. The authors express chimeric zinc-finger-DNA deaminase variants in bacteria and test the ability of the deaminases to revert an ACG to an ATG codon in GFP. They demonstrate that targeted deamination is most robust with AID and that the deamination frequency is specific to the target sequence, and there are very few non-specific mutations. The novel chimeric proteins were also demonstrated to function in human cells on chromosomally integrated substrates without significant associated toxicity. These experiments led the authors to propose that targeted DNA deamination by AID and related deaminases could be a convenient way to generate targeted mutations, especially in systems where DNA cleavages are toxic.

With the advent of CRISPR/Cas9, much effort has been invested of late in designing strategies for robust targeted DNA manipulation. Even the APOBEC/AID family members have been demonstrated to function in precise genome editing by David Liu and colleagues. But the field continues to evolve with more efficient targeting strategies. Thus, new techniques, especially those with fewer collateral damage, as described in this manuscript, are of much interest. However, there are several issues with this manuscript that should be resolved before publication.

1. The authors should include catalytically inactive AID in their reversion assays to be confident that the effect they see is indeed DNA deamination by AID. Several catalytic mutants of AID have been described by others.

Thank you for your comments. In Fig. 1d, 1e and 2b, we used ZF as control and we observed only background level of GFP signal compared to ZF-AID. The same phenomenon is observed with TALE-AID compared with the TALE only control. We agree with the referee that more experiments are helpful but we believe that our current data set is sufficient to indicate that that AID contributes to the enzymatic activities of the fusion protein

A minor point, the catalytic domain is not in the C-terminus of AID as the authors have indicated.

We changed the language to correct our mis-statement that the catalytic domain is in the C-terminus

“Based on available structures of APOBEC2, we tethered the ZF to the N-terminus of the deaminases to prevent steric hindrance to catalysis, separated by a four amino-acid linker (Fig. 1c).”

2. An important control that is missing throughout the manuscript is expression levels of the various chimeric molecules. There are no western blots for AID expression, and without this data it is impossible to be convinced that the differences in deamination activity is really due to functional modulation of the deaminase and/or targeting sequence.

Thank you for the comments. We agree that expression data is helpful for the fair comparison of different deaminases and can potentially help us increase the efficiency if we find that expression is suboptimal and can be increased. The main objective of this manuscript is to characterize the targeted deaminases system for efficiency and specificity. Our major discovery is that the processivity and the intrinsic DNA binding activity of deaminases can cause unintended mutagenesis in the genome independent of fused DNA domain binding activities. We believe that the expression level of ZF-AID is sufficient to reveal significant off-target activity of this enzyme. We hope that this point is clearer in the revised manuscript.

3. It would be important to test if the deamination target matches that of the deaminase used. AID and APOBEC3 for example have different target specificities. Can the ACG sequence used in the reversion assay be altered so that now APOBEC3G is a better deaminase for editing? This kind of data is important as it would allow one to customize deaminases for different target sequences.

This is an excellent point! Indeed most of the APOBEC proteins prefer the CC dinucleotides whereas APOBEC3B prefers TC. It is unfair to compare the activity of all APOBECs using the ACG to ATG conversion in the GFP reporter. Interestingly, Komor et al. suggests that targeted deaminases do not have strong preference at the on-target site due to local high enzyme concentration. Again, the goal of this manuscript is to explore the feasibility of targeted deaminases and examine their specificities, and we believe the activity of our ZF-AID fusion is sufficient for this goal.

In the revised manuscript, we clarified that “*We do not exclude the potential activity of other deaminase domains as ACG is not a preferred site for some of them*” and that “*In this study, we did not observe targeted genome editing activities of other deaminases (APOBEC1, 3F, 3G, Figure 1a), however, we did not exclude the possibility that expression level as well as sequence preference contributed to the negative signals. We look forward to future studies to explore the customized targeted deaminases for different target sequence.*”

4. The results generated with the C-terminal truncation mutation is interesting but it is important that this experiment is better controlled. The C-terminal deletion dramatically destabilizes AID, even though there is increased frequency of switch region mutation in B cells, and it is only under specific cases such as an ER-fused AID wherein the truncated protein is stabilized. Overall, this is a very complicated mutation. Thus, it is important to characterize at least the expression level of the protein in the context of the assay.

Thank you for the comment. We recognized the complication of this mutation and discussed it in the revised manuscript.

“*This mutation, despite the potential to destabilize the protein, is also expected to correctly localize the ZF-AID to the nucleus and decouple AID from the mismatch repair pathway.*”

The key message of our manuscript is that in an attempt to characterize the deaminase system in terms of efficiency and specificity, we found the specificity is not ideal to support the practical use of

this tool due to the AID's intrinsic semi-random mutagenesis potential. We look forward to future studies with further AID engineering to abolish its DNA binding affinity.

Reviewer #2 (Remarks to the Author):

The paper by Yang and coworkers examines cytidine deaminases conjugated with zinc finger (ZF) proteins or transcription activator-like effectors (TALE) as a new genome editing enzyme unit. The authors tested candidate deaminases for genome editing activity using ZF proteins or TALEs as the DNA binding module and optimized the conditions for better editing efficiency using reporter systems in *E. coli*. The authors then identified the off-target activity pattern of the deaminases and found increased global C-to-T background mutations. They also observed that these novel enzymes also function in human cells at an efficiency of 2.5%.

A similar but more comprehensive and potentially more significant study using Cas9 nickase combined with cytidine deaminase was published by David Liu's group in Nature April 20, 2016 [Komor et al. (ref. 30)]. The study under review is different from the Liu group's study in that the authors used ZF proteins or TALEs as the DNA binding module. However, both the broad range of C-to-T conversion near the binding sites and the low efficiency of the current enzyme limit the significance and potential applications of this enzyme for precise genome editing.

Major comments:

1. The low deaminase efficiency shown here clearly limits the significance of this study. The authors should seek more ways to improve the efficiency of targeted deaminases. Possible approaches may include, but are not restricted to, the following:

1.1. Has the binding of the relevant ZF proteins or TALEs to the target DNA been validated? The low deaminase activity might be due to improper binding of the enzyme to the target DNA sequence, which the authors should check.

Thank you for your comments. We added more references from our group and others who validated the efficiency of ZF and TALE binding. We also make this point more apparent in our revised manuscript.

"To test this, we first engineered targeted deaminases by fusing each candidate deaminases (APOBEC1, APOBEC3F, APOBEC3G (2K3A)16 and AID) with a sequence-specific ZF, which has proven to be effective by previous studies [R] (recognizing the 9bp DNA sequence 5'-GCCGCAGTG-3'17 (Fig. 1a)."

1.2. The authors of the current study tethered ZFs or TALEs to the N-termini of various deaminases and found that AID showed comparatively robust efficiency among those tested. The recent work by Komor et al. (ref. 30) examined deaminase activity in mammalian cells after tethering Cas9 to deaminase C-termini and found that rAPOBEC1 was the better genome editing deaminase. These different results might be caused by the different DNA binding units, ZF or TALE versus Cas9, used in the two studies and/or fusion of the DNA binding units to the N- versus C-termini. The authors could check whether changing either of these variables would enhance efficiency in their system.

Thank you for the comment. As we now mention in our updated Discussion section, we think that the difference in efficiency between our system and the study reported by Komor et al are also due to the strand displacement function of the Cas9-gRNA complex, which makes it easier for the deaminase to access its ssDNA substrate. This function cannot be achieved by either ZF or TALE.

Admittedly, further optimizations can increase the efficiency, but we hope that through our manuscript we make it clear that we are not trying to sell a tool for precise genome editing. Rather, our paper is an honest exploration of this alternative system to test its efficiency and specificity. We discovered that the deaminase intrinsic DNA binding activity and processivity are still preserved in the chimeric protein,

which can lead to global genome mutagenesis at the AID hotspot (WRC). We hope our study can invite future engineering to abolish the intrinsic activities of deaminases to make this tool usable.

1.3. Komor et al. (ref. 30) fused uracil DNA glycosylase inhibitor (UGI) to the C-terminus of Cas9 to increase efficiency, a possibility that should be tested in this study as well. Using mutant bacterial cell lines or perturbing cell behavior with a UGI inhibitor and MSH2 shRNA, as proposed by the authors, could significantly limit the application of the methods.

We agree that the fusion strategy is a better method to localize the activities of UGI. We contrast our system with Komor's system in the Discussion section of our revised manuscript.

2. In the 3rd paragraph of the section titled 'Specificity of targeted deaminases', the authors indicated that the activity window is +/- 15 bp. Actually, the chance of only one C occurring within 30 bp is very rare and this method would be difficult to apply for precise genome editing.

We totally agree. One point we are suggesting, which is novel in this study, is that off-target deamination may arise from the promiscuity of the linker as well as the processivity of the deaminase domain. In contrast to the optimism of the Komor et al paper, we would like to warn the audience about the potential off-target effects and tumorigenesis of the therapeutic application of this tool, and inspire further engineering to address the specificity issues. We have revised our abstract, introduction, and discussion significantly to strengthen this point.

3. The authors did not mention the distance between the ZF binding site and the target site in the main manuscript, although supplemental figure 1a and supplemental method 6 describe the distance as 9 bp. Because the distance could be critical for enzyme activity, it would be beneficial if the authors could explain the distance determination in a greater detail.

Thank you for your suggestion. We would like to reiterate that the goal of this manuscript is to optimize these tools to the extent that we can robustly characterize the off-target effects. Because the ZF-AID showed sufficient activity with the 9bp distance, we did not perform further optimizations as we did with the TALE-AID.

We have revised our manuscript to reflect this point.

"To determine editing efficiency in vivo, we integrated a single-copy GFP reporter into the E. coli genome19 (Fig. 1b and Supplementary Methods 2) in which the GFP is normally not expressed due to a 'broken' start codon ('ACG') and the ZF binding site is 9bp from the target "C" in the start codon."

4. After determining the importance of linker length using ZF-AID, the authors optimized TALE-AID activity by truncation of the TALE C-terminus, not by regulating linker length. This decision requires an explanation.

Thank you for your suggestion, we revised this point in our revised manuscript.

"Given the importance of the linker between the DNA-binding module with the deaminase and TALE-C terminus was engineered as a linker for many TALE fusion proteins, we proceeded to investigate if truncation of the 178aa20. C-terminus could increase TALE-AID activity (Fig. 2c)."

5. Figure 3c: ZF-AID showed off-target effects at sites >150bp away from its original target site. The 9 bp ZF binding sequence is somewhat short because such a site might occur more than once in a genome. If the authors would analyse the sequences around these unwanted mutations and in addition use longer ZF binding sequences, they might find the reason for these off-target effects.

We apologize for the confusion. In fact, we checked the pseudo-ZF binding sites from the WGS data and did not find off-target mutations. We hope that this point is clear in our revised manuscript.

"In addition, we did not find any off-target mutations at predicted ZF/TALE off-target sites. The fact that off-target mutations are enriched at WRC motifs - the canonical AID recognition sequence²⁴ - suggest that AID in the fusion protein still maintains its intrinsic DNA binding activities and contribute to the elevated mutagenesis in the genome. "

6. Figure 3c-f: The authors sequenced bacterial genomes to assess genome-wide off-target effects; however, they used only one uninduced control per experiment. Considering the variation between the two uninduced controls in e and f, we suggest testing at least two or three uninduced controls per experiment to allow more solid conclusions to be reached about genome-wide off-target effects.

Thank you for your comment. We did the Wilcoxon statistical analysis to compare between the induced strain and non-induced strain, and we found that the deamination efficiency is significantly different in the WRC domain. This phenomenon is reproducible with ZF-AID and TALE-AID.

7. There are quite a few typos and errors. The following are some of the examples.

7.1. In the [Specificity of targeted deaminases] section: Typographical errors in the sentence that starts with "In the GFP- population, the only variant..." *Fixed*

7.2. Figure 1b: "deamianse" should be changed to "deaminase". *Fixed*

7.3. Figure 2c: "NLS" should be changed to "Linker". *Fixed*

7.4. Supplemental figure 1a: Start codons of dimers should be corrected to ACG. *Fixed*

Thank you very much for the careful review. We have corrected all the aforementioned typos and errors in the revised manuscript.

Minor comments:

1) The species of the tested deaminases should be mentioned.

We have added the species of deaminases in the revised version.

"To test this, we first engineered targeted deaminases by fusing each candidate deaminases (human APOBEC1, APOBEC3F, APOBEC3G (2K3A)¹⁶ and AID) with a sequence-specific ZF, which has proven to be effective by previous studies (recognizing the 9bp DNA sequence 5'-GCCGCAGTG-3'¹⁷ (Fig. 1a)."

2) Some of the figure citations in the text do not match with what is shown in the figures. I would suggesting checking every figure citation.

Example: In the [Optimization of targeted deaminases] section, line 23: Fig 2e is cited, yet Fig 2 does not contain a section "e".

Thank you very much. We have revised carefully all of our figures, and have made appropriate edits in the revised manuscript.

3) In the [Optimization of targeted deaminases] section, line 3: "a longer linker length improved editing frequencies" is an ambiguous explanation of the data. An enzyme containing an 11-amino acid linker showed lower activity than one with an 8-amino acid linker.

We updated the description in the revised manuscript.

"We next conducted structural optimization of the targeted deaminases by varying linker lengths and sequence compositions^{21, 22} (Fig. 2a). While tested variants all led to robust GFP rescue, with ZF-8-aa-AID achieving 7.5% GFP+ frequency after 10 hours (Fig. 2b) and 13% after 30 hours of induction (Supplementary Fig. 2a)."

4) In the [Optimization of targeted deaminases] section, lines 10-11: "C-terminus of TALE" would be more accurate than "C-terminus"

We have updated the manuscript by incorporating this suggestion.

“Given the importance of the linker between the DNA-binding module with the deaminase and TALE-C terminus was engineered as a linker for many TALE fusion proteins, we proceeded to investigate if truncation of the 178aa20. C-terminus of TALE could increase TALE-AID activity (Fig. 2c).”

5) In the [Specificity of targeted deaminases] section: The authors claimed that TALE-C3-AID showed strong sequence specificity for the first 8 bp. When I counted the bases in this region in the figure, it is 9 bp rather than 8 bp. The authors should check the number again.

Thank you for your careful review. We have updated the correct number in the manuscript;

“We next investigated the specificity of TALE-AID by individually varying each nucleotide in the TALE recognition site to the second most preferred base¹ for that position (Fig. 3b), and tested TALE-AID targeting efficiency on individual reporters respectively. Interestingly, TALE-C3-AID, which was designed to recognize a 14bp sequence, showed strong sequence specificity only for the first 9bp proximal to the target site (5' CTTCTTCCC 3' in the TALE recognition site).”

6) Introduction: One of the limitations of homology directed repair is that it is restricted to G2/S phase and thus barely applicable to cells with a dormant cell cycle. Adding this point will strengthen the manuscript.

Thank you for your suggestion. We had similar thoughts but realized that replication is needed to resolve the U:G → A:T. However, if uracil is not repaired and is on the template strand of RNA transcription, RNA polymerase II is able to use uracil as a substrate and introduce mutations into the RNA sequence. We elaborated that in the Discussion Section of our revised manuscript.

“Moreover, One of the limitations of homology directed repair is that it is restricted to G2/S phase and thus barely applicable to cells with a dormant cell cycle. Targeted deaminases can potentially bypass such a problem by introducing C-to-U mutations on the template strand of dsDNA, which can be recognized by RNA polymerase II to produce RNA with intended mutations independent of cell cycle.”

REVIEWERS' COMMENTS:

Reviewer #1 (Remarks to the Author):

Most of the issues raised by the reviewers have been adequately addressed.

Reviewer #2 (Remarks to the Author):

I do not have any more concerns about this manuscript.